# Enhanced M2 Polarization of Retinal Microglia in Streptozotocin-Induced Diabetic Mice upon Autoimmune Stimulation [note 1]

**DOI:** 10.3390/biomedicines13092049

**Published:** 2025-08-22

**Authors:** Yoshiaki Nishio, Hideaki Someya, Kozo Harimoto, Tomohito Sato, Masataka Ito, Masaru Takeuchi

**Affiliations:** 1Department of Ophthalmology, National Defense Medical College, Tokorozawa 359-8513, Japan; cln349@ndmc.ac.jp (Y.N.); fwgi0461@ndmc.ac.jp (K.H.); dr21043@ndmc.ac.jp (T.S.); 2Department of Developmental Anatomy and Regenerative Biology, National Defense Medical College, Tokorozawa 359-8513, Japan; masataka@ndmc.ac.jp

**Keywords:** experimental autoimmune uveoretinitis, diabetes, microglia, autoimmune uveitis, autoimmune disease

## Abstract

**Background:** This study aimed to investigate the impact of the diabetic environment on the development of experimental autoimmune uveoretinitis (EAU) and the activation status of microglia in the eye. **Methods:** EAU was induced in wild-type (WT) and streptozotocin (STZ)-induced diabetic mice (STZ-EAU mice). Disease severity was assessed using funduscopy, optical coherence tomography (OCT), and histopathological analysis. The proportions of Th1, Th17, and regulatory T cells in the spleen were analyzed by flow cytometry. Retinal microglia were quantified using immunohistochemistry. To further characterize retinal cell populations and gene expression profiles, single-cell RNA sequencing (scRNA-seq) was performed. **Results:** STZ-EAU mice exhibited significant reductions in both the incidence and severity of EAU compared with WT-EAU mice. These were accompanied by a decreased proportion of Th1 cells, which are crucial for EAU pathogenesis, in the spleens of STZ-EAU mice. Retinal microglial accumulation was markedly reduced in STZ-EAU mice compared with WT-EAU mice. scRNA-seq analysis revealed a significant change in the microglial phenotype in STZ-EAU mice, characterized by decreased expression of MHC class I/II and the suppression of antigen presentation signaling pathways. Activated microglia in STZ-EAU mice showed reduced gene expression of M1 markers (*CD68*, *CD74*, and *IL1B*) and increased gene expression of M2 markers (*MSR1*, *CD163*, and *MRC1*), suggesting a shift toward an anti-inflammatory M2 phenotype. **Conclusions:** EAU is suppressed in STZ-induced diabetic mice, likely due to alterations in microglial polarization toward an M2 phenotype. These results suggest a decrease in T cell responses to pathogens in a diabetic environment, which could be one of the underlying factors for the increased susceptibility to infection in diabetic patients. Inhibiting the M2 polarization of microglia may reduce the susceptibility to infection in patients with diabetes.

## 1. Introduction

Diabetic retinopathy (DR) is a sight-threatening complication of diabetes mellitus (DM) that damages both the neurons and the microvasculature of the retina [1]. While initially considered solely a microvascular disease, characterized by features such as microaneurysms and acellular capillaries [2], increasing evidence highlights the crucial role of inflammation in the progression of DR [3,4,5,6]. This understanding has prompted a shift in focus toward the interplay between inflammation and immune dysregulation in DR pathogenesis. Within the retina, activated microglia, resident immune cells located in the plexiform layers, drive these inflammatory reactions [2,7,8]. Microglia typically act as sentinels, transitioning to an activated state to neutralize threats and maintain tissue homeostasis upon encountering danger signals [9]. However, persistent stress can push microglia into a hyperreactive state characterized by the excessive release of pro-inflammatory mediators such as cytokines and chemokines [10].

Autoimmune diseases arise from abnormal immune responses directed against self-antigens as a result of immune system dysregulation [11]. Experimental autoimmune uveoretinitis (EAU), induced by the immunization of experimental animals with the retinal antigen interphotoreceptor retinoid-binding protein (IRBP), serves as an animal model of human endogenous uveitis [12]. Its pathogenesis involves an immune response mediated by antigen-presenting cells and T cells that specifically recognize retinal antigens [13,14,15]. Retinal microglia are involved in the development of EAU by acting as retina-specific antigen-presenting cells [15]. Upon EAU induction, retinal microglia become activated before cellular infiltration occurs around retinal vessels and play a crucial role in facilitating the infiltration of antigen-specific T cells into the retina [16,17]. Agents that suppress microglia have demonstrated efficacy in inhibiting EAU. Dexamethasone suppresses the increase in microglial cell density during EAU development and prevents microglial infiltration into the retinal inner nuclear layer (INL) and outer nuclear layer (ONL) [18,19]. Inhibition of galectin-3 reduces microglial production of inflammatory cytokines and mitigates the severity of EAU [20]. Similarly, apigenin inhibits microglial activation and suppresses EAU [21]. These findings underscore the crucial role of retinal microglia in the pathogenesis of autoimmune uveitis.

Since inflammation resulting from microglial activation in the retina contributes to the development and progression of DR in individuals with diabetes [22], we hypothesized that EAU would be exacerbated in diabetic mice. Unexpectedly, EAU was suppressed in Ins2^Akita^ (Akita) mice, which carry a spontaneous mutation in the insulin 2 gene, resulting in severe insulin-dependent diabetes beginning at 3 to 4 weeks of age [23]. However, it is possible that the genetic mutation in Akita mice could impact the expression of genes beyond Ins2, potentially including those related to immune responses. Therefore, we considered it essential to verify this finding in mice without genetic mutations.

In the present study, to investigate whether EAU is suppressed under diabetic conditions in animals without genetic mutations, we induced diabetes by administering streptozotocin (STZ) to wild-type C57BL/6 mice and examined the occurrence of EAU. Furthermore, we aimed to elucidate the role of retinal microglia in the pathogenesis of EAU within the diabetic environment. To achieve this, we performed single-cell RNA sequencing (scRNA-seq) of retinal cells to analyze the immunological characteristics of microglia, including their intracellular signaling pathways, differentiation states, and interactions with other retinal cell populations. This analysis aims to provide insights into how diabetes-induced alterations in microglial function may contribute to the modulation of EAU.

## 2. Materials and Methods

### 2.1. Animals

All C57BL/6J mice used in this study were purchased from Japan SLC Inc. (Shizuoka, Japan). All mice were housed in the Center for Laboratory Animal Science of the National Defense Medical College under specific pathogen-free conditions with a regular light/dark cycle (14 h of light and 10 h of darkness per day) and access to food and water ad libitum. A one-week acclimation period was provided for all mice before the experiments. The animal experiment protocols were reviewed and approved by the Animal Ethics Committee of the National Defense Medical College (approval number: 23067-2), and the procedures were carried out according to the Association for Research in Vision and Ophthalmology Statement for the Use of Animals in Ophthalmic and Vision Research. According to the animal experiment protocols, abnormal mice showing rapid weight loss (>10%) or decreased activity were excluded from the experiment. All anesthesia was performed by the intraperitoneal administration of 0.1 mL of 0.75 mg/kg medetomidine, 4.0 mg/kg midazolam, and 5.0 mg/kg butorphanol per 10 g of body weight. After the experiment, mice were promptly awakened by the intraperitoneal administration of 0.1 mL of 0.75 mg/kg atipamezole, a medetomidine antagonist, per 10 g of body weight. This study was conducted using three groups: C57BL/6J wild-type mice (WT), WT mice with induced EAU (WT-EAU), and immunized STZ-induced diabetic mice with EAU (STZ-EAU).

### 2.2. Creation of STZ-Induced Diabetic Mice

STZ (Wako, Osaka, Japan)-induced diabetic mice were generated as previously described [24]. Briefly, 5-week-old female C57BL/6J mice were intraperitoneally injected with 60 mg/kg STZ dissolved in PBS for five consecutive days. Mouse blood glucose levels were measured weekly using a LAB Gluco glucometer (Research & Innovation Japan Inc., Chiba, Japan) from a small amount of blood collected by excising the tip of the mouse tail. Mice with blood glucose levels of 300 mg/dL or higher four weeks after STZ administration were used in the experiments. A comparison of blood glucose levels and body weight between STZ-induced diabetic mice and WT mice is shown in Appendix A.

### 2.3. EAU Induction

EAU mice were generated as previously described, with a minor modification [23]. Briefly, WT mice and STZ-induced DM mice were deeply anesthetized and immunized subcutaneously at two sites in the lumbar region with 200 µg of human IRBP 1-20 (hIRBP-p) emulsified in 0.2 mL of complete Freund’s adjuvant (Difco, Detroit, MI, USA) containing 1 mg of *Mycobacterium tuberculosis* strain H37Ra (Difco), and received an intraperitoneal injection of 0.5 μg of pertussis toxin (Sigma Aldrich, St. Louis, MO, USA) dissolved in 0.2 mL of PBS.

### 2.4. EAU Scoring

Funduscopy and optical coherence tomography (OCT) were performed on mice under anesthesia at 21 days post-immunization using a Micron IV fundus camera with MICRON Image-Guided OCT (Phoenix Research Labs, Pleasanton, CA, USA) to assess the severity of EAU. The severity of EAU was also evaluated histopathologically using tissue sections prepared from eyeballs collected at 21 days post-immunization. Individual EAU scores were graded from 0 to 4 based on a previously reported scale [25]. These EAU scores were evaluated in histopathology sections by two ophthalmologists who were blinded to the sample groups.

### 2.5. Flow Cytometry

After euthanasia by cervical dislocation, spleens were harvested from WT-EAU and STZ-EAU mice at 21 days post-immunization and then homogenized. Subsequently, red blood cell lysis was performed using Red Blood Cell Lysis Buffer (Roche, Basel, Switzerland) to obtain splenocytes. All splenocytes were pretreated with an anti-mouse CD16/CD32 antibody (eBioscience, San Diego, CA, USA) to block Fc receptors before staining, and subsequent staining was performed. To detect Th1 and Th17 cells, cells were resuspended in complete RPMI1640 medium (Wako) supplemented with 10% fetal bovine serum (Gibco, Waltham, MA, USA), 1 mM sodium pyruvate (Gibco), 1% nonessential amino acids (Gibco), 100 U/mL penicillin and 100 µg/mL streptomycin (Gibco), and 50 µM 2-mercaptoethanol (Wako). The cells were then stimulated with 50 ng/mL phorbol 12-myristate 13-acetate (PMA; Sigma-Aldrich, St. Louis, MO, USA) and 0.5 µg/mL ionomycin (Sigma-Aldrich) in the presence of 10 µg/mL brefeldin A (Sigma-Aldrich) for 4 h at 37 °C. After stimulation, surface markers were stained, followed by fixation and permeabilization using a permeabilization/fixation buffer (Becton Dickinson (BD) Bioscience, Franklin Lakes, NJ, USA) according to the manufacturer’s instructions. Intracellular staining was then performed for IFN-γ or IL-17A. For the detection of regulatory T cells (Tregs), cells were first stained for surface markers, then fixed and permeabilized using the Foxp3/Transcription Factor Staining Buffer Set (eBioscience) according to the manufacturer’s instructions, followed by staining for the nuclear marker Foxp3. Antibodies used for staining in each experiment are listed in Appendix A. Following staining, cytometry data were acquired by a FACSCanto II (BD), and analyzed with BD FACSDiva™ software. Version 8.0.1 (BD). The gating strategies are shown in Appendix A.

### 2.6. Histological and Immunohistochemical Analyses

Mice were deeply anesthetized and perfused with 4% paraformaldehyde in PBS. After euthanasia, eyeballs were enucleated and post-fixed overnight with 4% paraformaldehyde in PBS. The tissues were then embedded in paraffin. Paraffin sections (5 μm) were cut and deparaffinized. Hematoxylin and eosin (H&E) staining was performed using an automated slide stainer (Tissue-Tek Prisma^®^, Sakura Finetek Japan, Tokyo, Japan) with hematoxylin 3G and eosin solutions (Sakura Finetek Japan). For immunohistochemistry, after sections were deparaffinized with xylene, antigen retrieval was performed by incubating sections in antigen unmasking solution (Vector Laboratories, Newark, CA, USA) and heating them in an autoclave (121 °C) for 5 min. Endogenous peroxidases were blocked with a peroxidase-blocking reagent (Dako, Santa Clara, CA, USA). Sections were then incubated with 2.5% normal horse serum, followed by an incubation with an anti-Iba-1 antibody (Wako) diluted 1:1000 in antibody diluent (Dako). After the incubation with the primary antibody, the sections were incubated with a peroxidase-conjugated secondary antibody (Vector Laboratories). The reaction product was visualized with 3,3′-diaminobenzidine (DAB) and counterstained with hematoxylin. The number of Iba-1-positive cells in each retinal layer was measured using the average number of positive cells in two high-power fields per slide, and these counts were evaluated in histopathology sections by two ophthalmologists who were blinded to the sample groups.

### 2.7. Single-Cell RNA Sequencing (scRNA-Seq)

#### 2.7.1. Retinal Single-Cell Preparation

Retinal single cells were prepared as described by Licheng Sun et al. [26]. Briefly, five mice per group (WT, WT-EAU, and STZ-EAU) at 21 days post-immunization were deeply anesthetized. Under anesthesia, retinal perfusion with 50 mL of PBS and exsanguination were performed for euthanasia. The eyeballs were then enucleated, and the retinas from each group of five mice were pooled and carefully dissected. Retinas were washed with PBS and treated with 500 µL of a 2 U papain solution containing 0.5 mM EDTA and 2.5 mM L-cysteine for 10 min at 37 °C. The enzyme reaction was quenched by adding 750 µL of AMES medium containing 10% FBS. Subsequently, 7.5 U of DNase I was added and incubated for 5 min at 37 °C. Single-cell suspensions of retinas were prepared by gentle pipetting.

#### 2.7.2. Rod Cell Depletion

Rod cells constitute the majority of retinal cells [27,28]. To reduce the proportion of rod cells in the retinal cell suspension, magnetic bead cell separation was performed using MACS Separator cell isolation system (Miltenyi Biotec, Bergisch Gladbach, Germany) with an anti-mouse CD73 antibody (BD Biosciences, Franklin Lakes, NJ, USA) and anti-rat IgG microbeads (Miltenyi Biotec), following the manufacturer’s instructions.

#### 2.7.3. scRNA-Seq Data Acquisition

The retinal single-cell suspension was loaded onto Chromium microfluidic chips with 3′ chemistry and individual cells were barcoded using a 10× Chromium Controller (10× Genomics, Pleasanton, CA, USA). After barcoding, the RNA from the isolated cells was reverse transcribed, and sequencing libraries were prepared using the Chromium Single Cell 3′ Reagent Kit v2 (10× Genomics), following the manufacturer’s instructions. Sequencing was performed on the NovaSeq6000 platform (Illumina, San Diego, CA, USA) in accordance with the manufacturer’s instructions. The resulting scRNA-seq reads were processed using Cell Ranger software v7.0.0 (10× Genomics) with the default settings for each sample independently.

#### 2.7.4. scRNA-Seq Data Analysis

Processed scRNA-seq data were analyzed by BBrowserX (BioTuring, San Diego, CA, USA). Genes with a *p*-value less than 0.05 based on the Venice test and |Log2 fold change| > 0.25 were considered differentially expressed genes (DEGs) [29]. Gene Ontology biological process (GOBP) enrichment analysis was conducted based on the Molecular Signatures Database (MSigDB) to identify significantly enriched GOBP terms among DEGs between groups.

### 2.8. Statistical Analysis

Statistical analyses were performed using JMP pro 15 (SAS Institute, Cary, NC, USA). Categorical data were compared using Fisher’s exact test, while nonparametric data between two groups were compared using the Wilcoxon rank-sum test. For comparison of three or more groups, the Steel–Dwass test was used. *p* values less than 0.05 were considered significant.

## 3. Results

### 3.1. EAU Development Is Suppressed Under STZ-Induced Diabetic Conditions

We first examined the incidence and severity of EAU in STZ-induced diabetic mice. Funduscopic examination revealed that eight of ten WT mice (80%) developed EAU at day 14 post-immunization, with a median clinical score of 1.00 (interquartile range (IQR): 0.38–1.25), while none of the STZ-induced diabetic mice developed EAU (Figure 1A). At day 21 post-immunization, one of ten STZ-induced diabetic mice (10%) developed EAU, with a median clinical score of 0.00 (IQR: 0.00–0.00). This was significantly lower than in WT mice, in which the incidence was nine of ten (90%) and the median clinical score was 1.00 (IQR: 0.88–2.00). Fundus color photographs of the most severe EAU cases in both WT and STZ-induced diabetic mice are shown in Figure 1B. On day 14 post-immunization, WT mice exhibited retinal vasculitis with vessel sheathing and surrounding exudates. By day 21, vasculitis had subsided. However, exudates were still observed across the retina. In contrast, STZ-induced diabetic mice showed no inflammatory signs on day 14. While only low-grade retinal vasculitis was observed on day 21, exudates were absent. These results demonstrate that EAU development is markedly suppressed in STZ-induced diabetic mice.

We also evaluated EAU using OCT and histopathological analyses. At day 21 post-immunization, the median OCT score was significantly lower in STZ-EAU mice (0.00, IQR: 0.00–0.00) than in WT-EAU mice (1.00, IQR: 1.00–2.00) (Figure 2A). Representative OCT images in each group showed that infiltrating cells in the vitreous cavity (white arrow) and the disruption of retinal layers (red arrow) were substantially more severe in WT-EAU mice than in STZ-EAU mice (Figure 2B). The histopathological evaluation also revealed that the median histological score at day 21 post-immunization was significantly lower in the STZ-EAU group (0.00, IQR: 0.00–0.00) compared to the WT-EAU group (1.00, IQR: 0.50–1.25) (Figure 2C). Representative histological sections from each group showed infiltrating cells (black arrow), vasculitis (blue arrow), and the destruction of retinal layers (red arrow) in WT-EAU mice, whereas STZ-EAU mice exhibited only a few infiltrating cells near the optic nerve head with minimal vasculitis and retinal destruction (Figure 2D).

### 3.2. Decreased Th1 Cell Proportion in Diabetic Mice

Since EAU development involves Th1 and Th17 cell differentiation, which are crucial for the pathogenesis of EAU, while Tregs suppress EAU development, we analyzed splenic CD4^+^ T cells, including Th1 and Th17 cells, in WT-EAU and STZ-EAU mice (WT and STZ-induced diabetic mice immunized with hIRBP-p) using flow cytometry. The proportion of naïve T cells, characterized by the absence of activation and marked by CD44^−^ CD62L^+^ expression, was comparable between WT-EAU (61.3%, IQR: 57.4–64.5) and STZ-EAU (64.1%, IQR: 60.5–71.8) mice (Figure 3A). Similarly, the proportion of activated T cells, indicated by CD44^+^ CD62L^−^ expression, showed no significant difference between WT-EAU (29.9%, IQR: 28.0–31.2) and STZ-EAU (29.3%, IQR: 22.1–32.9) mice (Figure 3A). These results suggest that overall T cell activation in EAU proceeds similarly in both WT and STZ-induced diabetic mice. However, the proportion of Th1 cells (CD4^+^ IFN-γ^+^ T cells) was significantly lower in STZ-EAU mice (6.2%, IQR: 4.6–7.0) compared to WT-EAU mice (8.1%, IQR: 7.7–9.6) (Figure 3B), whereas the proportion of Th17 cells (CD4^+^ IL-17^+^ T cells) showed no significant difference between the two groups (1.5%, IQR: 1.1–2.8 in WT-EAU mice and 1.0%, IQR: 0.6–1.9 in STZ-EAU mice) (Figure 3C). The proportion of Tregs (CD4^+^ Foxp3^+^ T cells) did not differ significantly between WT-EAU mice (2.7%, IQR: 2.1–2.9) and STZ-EAU mice (3.6%, IQR: 2.6–5.0) (Figure 3D).

### 3.3. Reduction in the Number of Microglia in Retina of the STZ-Induced Diabetic Mice with Autoimmune Uveitis

Retinal microglia are known to contribute to the inflammatory processes that drive EAU [17,18,19,20,21]. To investigate the impact of diabetes on retinal microglia in EAU, we quantified microglia in each retinal layer of WT, WT-EAU, and STZ-EAU mice. Compared to WT mice, WT-EAU mice showed a significant increase in the number of retinal microglia in the NFL, IPL, OPL, ONL, and RCL (Figure 4A). However, in STZ-EAU mice, a significant increase in retinal microglia compared to WT mice was observed only in the RCL (Figure 4A). The number of microglia in all retinal layers was lower in STZ-EAU mice than in WT-EAU mice, with significant reductions in the NFL, IPL, OPL, and ONL. Representative histopathological sections of retinal microglia in each group are shown in Figure 4B.

### 3.4. Transcriptome Atlas of Diabetic Retinal Cells in Autoimmune Uveitis

To comprehensively analyze retinal cells, we generated a transcriptome atlas of retinal cells in WT, WT-EAU, and STZ-EAU mice using scRNA-seq. scRNA-seq data from retinal cells of the three groups were integrated, and after creating a t-SNE plot, nine clusters were identified through cluster analysis (Figure 5A). Furthermore, each cluster was annotated by analyzing gene expression patterns characteristic of each retinal cell type (Figure 5B). The top 10 genes characteristically expressed in each cluster were selected and displayed in a heatmap (Figure 5C). The list of these characteristic genes is provided in Appendix A. The proportion of the nine classified retinal cell types in each group is shown in Figure 5D. Consistent with the immunohistochemical findings, the proportion of retinal microglia was increased in WT-EAU mice (9.8%) and STZ-EAU mice (2.0%) compared with WT mice (1.1%), with a markedly smaller increase in STZ-EAU mice (Figure 5D). Additionally, the top 10 genes that were characteristically expressed in WT, WT-EAU, or STZ-EAU mice were compared within each retinal cell cluster using a heatmap (Figure 5E). Three types of gene expression patterns were detected in the heatmap: the top panel shows genes decreased by EAU, the middle panel shows genes increased in STZ-EAU mice, and the bottom panel shows genes increased in WT-EAU mice but not in STZ-EAU mice. Therefore, the bottom gene group comprises genes associated with EAU and modulated by diabetes, with microglia exhibiting the highest expression of these genes among the retinal cells (Figure 5E). The characteristic genes for each group are listed in Appendix A.

### 3.5. M2 Polarization of Retinal Microglia in Diabetic Mice with Autoimmune Uveitis

We focused on the microglia cluster for further analysis. Differentially expressed gene (DEG) analysis was performed using microglia from the comparison of the STZ-EAU and WT-EAU groups, and the differences in gene expression were visualized in a volcano plot (Figure 6A). The most significantly downregulated DEGs in the STZ-EAU group compared to the WT-EAU group were MHC class I molecules (*H2-Q7*, *H2-K1*, and *H2-D1*) and MHC class II molecules (*H2-AA* and *H2-EB1*), as well as *CD74*, a molecular chaperone critical for antigen presentation via MHC class II [30]. Additionally, *P2RY12* and *SIGLECH*, which are specifically expressed in microglia and downregulated upon activation [31,32,33], were detected as upregulated DEGs in STZ-EAU mice. Gene Ontology (GO) analysis was performed using DEGs from the comparison of the STZ-EAU and WT-EAU groups. The top 10 GO terms identified were enriched for antigen processing and presentation, which were downregulated in STZ-EAU mice (Figure 6B). Subsequently, sub-cluster analysis of the microglia cluster identified five distinct sub-clusters (Figure 6C). Sub-clusters 1, 4, and 5 were predominantly composed of cells from WT-EAU mice, while sub-cluster 2 contained cells from both WT and STZ-EAU mice. Sub-cluster 3 consisted of the highest number of cells from WT-EAU mice (129 cells), followed by STZ-EAU mice (29 cells) and WT mice (8 cells) (Figure 6D). *AIF1* is a gene expressed in both retinal microglia and infiltrating macrophages, while *CX3CR1*, *P2RY12*, *TMEM119*, and *SIGLECH* are microglia-specific genes [31,32,33]. Notably, P2RY12, TMEM119, and SIGLECH serve as markers for quiescent microglia. CD40, CD80, and CD86, co-stimulatory molecules essential for T cell activation, are highly expressed on M1 microglia [34]. The analysis of microglia-specific gene expression revealed that all sub-clusters were *AIF1*-positive (Figure 6E). However, based on the expression of *CX3CR1*, *P2RY12*, *TMEM119*, and *SIGLECH*, sub-clusters 1 and 2 appear to consist of quiescent microglia, sub-cluster 3 of activated microglia, and sub-clusters 4 and 5 of infiltrating macrophages. MHC class II molecules in antigen-presenting cells in the retina are upregulated with the onset of EAU [35]. The activation states of microglia and infiltrating macrophages were clearly reflected by MHC molecule expression, with lower expression of *H2-AA*, *H2-EB1*, and *H2-AB1* in sub-clusters 1 and 2, and significantly higher expression in sub-cluster 3 (Appendix A).

An analysis of the expression of M1 and M2 marker genes in sub-cluster 3 (activated microglia) revealed that the expression of *CD68*, *CD74*, and *IL1B* was upregulated in sub-cluster 3 microglia from WT-EAU mice but significantly downregulated in sub-cluster 3 microglia from STZ-EAU mice (Figure 6F). The expression of the M2 marker genes *MSR1*, *CD163*, and *MRC1* was localized to sub-cluster 3 microglia, with lower expression in sub-cluster 3 microglia from WT-EAU mice and higher expression in sub-cluster 3 microglia from STZ-EAU mice (Figure 6G).

GO analysis of the STZ-EAU and WT-EAU groups within sub-cluster 3 revealed the enrichment of GO terms related to neuronal functions, including “sympathetic neuron axon guidance” and “trigeminal nerve morphogenesis” (Figure 6H).

## 4. Discussion

The current study demonstrates that EAU is suppressed in STZ-induced diabetic mice compared to WT mice. This observation is consistent with our prior findings in Akita mice, suggesting that a diabetic environment attenuates EAU progression.

In our previous study, immunization with hIRBP-p led to a decrease in naïve T cells (CD44^−^CD62L^+^) and an increase in activated T cells (CD44^+^CD62L^−^) in Akita mice, and the same changes were observed in WT mice. However, Th1 differentiation was significantly lower in Akita-EAU mice compared to WT-EAU mice, whereas Th17 and Treg differentiation was preserved [23]. These results were compatible with the present study using STZ-induced diabetic mice. Given that AP-1 signaling was downregulated in Th cells of Akita-EAU mice [23], a similar suppressive effect on Th cells may also occur in STZ-EAU mice.

Although retinal antigen-specific T cells drive EAU pathogenesis, retinal microglia play a crucial role in disease development [17]. Retinal microglia increased in both WT-EAU and STZ-EAU mice following hIRBP-p immunization. However, the increase was significantly lower in STZ-EAU mice compared to WT-EAU mice. Notably, microglia numbers in the NFL, IPL, OPL, and ONL were significantly lower in STZ-EAU mice than in WT-EAU mice. scRNA-seq analysis supported these findings, revealing a smaller increase in retinal microglia in STZ-EAU mice compared with WT-EAU mice. Heatmap analysis further identified genes specifically upregulated in STZ-EAU retinal microglia, indicating that the reduced microglial accumulation in STZ-EAU mice is not merely a consequence of suppressed EAU but may also involve active regulatory mechanisms.

Sub-cluster analysis of retinal microglia revealed two quiescent microglia clusters (sub-clusters 1 and 2). Sub-cluster 1 was composed primarily of microglia from WT-EAU mice, while sub-cluster 2 mainly consisted of microglia from WT and STZ-EAU mice. Interestingly, based on the gene expression intensities of the quiescent microglial markers *P2RY12*, *TMEM119*, and *SIGLECH*, sub-cluster 1 appears to be in transition toward an activated state. However, STZ-EAU-derived microglia in this cluster were rare. Sub-clusters 4 and 5, representing infiltrating macrophages, were predominantly derived from WT-EAU mice, which may correlate with EAU severity and explain the reduced microglial accumulation in STZ-EAU retinas. Sub-cluster 3, comprising activated microglia, contained mostly microglia from WT-EAU mice, followed by STZ-EAU mice and then WT mice.

Retinal microglia normally contribute to maintaining tissue homeostasis in a quiescent state [36]. In DR and other inflammatory conditions such as uveitis, hyperglycemia directly activate microglia via advanced glycation end-product (AGE) formation, oxidative stress, and inflammatory signaling pathways [8,37,38,39], resulting in the release of pro-inflammatory cytokines, chemokines, and reactive oxygen species (ROS). Activated microglia can be broadly classified into two phenotypes: M1 microglia, which are pro-inflammatory and release cytokines such as TNF-α, IL-1β, and IL-6 that contribute to chronic inflammation and further retinal damage, and M2 microglia, which are anti-inflammatory and release cytokines like IL-10 and TGF-β that promote tissue repair and the resolution of inflammation [40]. In the pathogenesis of EAU, M1 microglia initiate and exacerbate retinal inflammation, adopting an amoeboid shape and migrating to the ONL and subretinal space [21]. Unexpectedly, however, an analysis of M1 (*CD68*, *CD74*, and *IL1B*) and M2 (*MSR1*, *CD163*, and *MRC1*) marker gene expression in activated microglia within sub-cluster 3 revealed a distinct pattern. The expression of M1 markers was higher in WT-EAU mice compared to STZ-EAU mice. Conversely, the expression of M2 markers was elevated in STZ-EAU mice compared to WT-EAU mice. This observation is consistent with the results of the DEG analysis, which demonstrated reduced expression of MHC genes (*H2-AA*, *H2-EB1*, *H2-D1*, *H2-K1*, and *H2-q7*) in STZ-EAU mice compared to WT-EAU mice. GO analysis of these DEGs further indicated a decrease in reactions related to antigen presentation and immune response. Diabetes impairs innate and adaptive immune function. Neutrophils from diabetic patients show reduced ROS production [41], disrupting degranulation [42], and impaired neutrophil extracellular trap formation [43]. Similarly, natural killer cells exhibit an impaired degranulation function in the diabetic environment [44]. Diabetes also impairs macrophage function, as peripheral blood monocytes from type 2 diabetes patients show a reduced phagocytic capacity [45,46], and bone marrow-derived macrophages from diabetic mice exhibit impaired phagocytosis, H_2_O_2_ production, and antimicrobial activity in high-glucose media [47,48]. Furthermore, Samuel Philip Nobs et al. demonstrated that hyperglycemia-induced glucose metabolic abnormalities lead to the dysfunction of adaptive immune responses to infections by causing decreased expression of co-stimulatory molecules in lung-resident dendritic cells and impaired T cell priming [49]. These suppressive effects of hyperglycemia on immune cells collectively weaken acquired immunity, contributing to increased susceptibility to infections.

Focusing on the M1/M2 polarization of macrophages and microglia in a hyperglycemic environment, peripheral blood monocytes of type 2 diabetes patients show a higher proportion of the M1 phenotype and a lower proportion of the M2 phenotype compared to controls [50]. Consistent with this trend, retinal microglia in STZ-induced diabetic rats display a time-dependent increase in M1 polarization and a decrease in M2 polarization after diabetes onset [51]. Interestingly, short-term exposure to hyperglycemia induces an M2 phenotype in microglia, whereas prolonged exposure shifts them toward an M1 phenotype, suggesting that a certain proportion of M2 microglia persists even under chronic hyperglycemia [52]. In proliferative diabetic retinopathy (PDR), a severe condition leading to vision loss, M2 microglia contribute to excessive angiogenesis through the production of TGF-β and VEGF [40,53]. In addition, M2 microglia are related to fibrotic scar formation [54]. M2 microglia are activated by Th2-related cytokines [36], and we have reported that the Th2-related cytokine IL-4 is produced in the vitreous fluid of PDR patients [5,6], suggesting that activated M2 microglia exacerbate PDR by promoting abnormal angiogenesis and fibrosis in the eye. Taken together, these findings indicate that inhibiting M2 microglia polarization could be an effective therapeutic strategy for DR.

Jiangyi Liu et al. demonstrated that CD74 and CCL5 expressed by retinal microglia in EAU contribute to disease pathogenesis [55]. CCL5 is a chemokine that induces the migration of macrophages and T cells to sites of inflammation [56]. Since retinal microglia infiltrate the photoreceptor layer prior to T cell infiltration, their activation is critical for recruiting immune cells into the eye. scRNA-seq analysis revealed that *CCL5* was the most upregulated molecule in retinal microglia during EAU. In contrast, DEG analysis comparing microglia from STZ-EAU and WT-EAU mice showed reduced expression of both *CD74* and *CCL5* in STZ-EAU mice, suggesting that they may be less capable of recruiting the immune cells required for EAU development. The downregulation of CCL5 promotes the M2 polarization of macrophages [57], and in a diabetic environment, reduced CCL5 production by retinal microglia may further promote M2 polarization. This shift toward an M2 phenotype could suppress T cell migration and activation, thereby contributing to the inhibition of EAU.

Nevertheless, further research is needed to elucidate the precise mechanisms underlying the observed shift toward an M2 phenotype in our STZ-EAU model. Investigating the roles of specific signaling pathways and metabolic factors in this process could provide valuable insights into the pathogenesis of DR and EAU and identify potential therapeutic targets.

## 5. Conclusions

Our present study demonstrates that the development of EAU is suppressed in STZ-induced diabetic mice, which is likely attributable to alterations in microglial activation and polarization. The diabetic environment appears to promote a shift toward an M2 microglial phenotype, resulting in reduced antigen presentation and potentially diminished recruitment of inflammatory cells to the retina. These results provide valuable insights into the complex interplay between diabetes and suppressed antigen-specific immune responses in the retina, which may be one of the underlying factors contributing to increased susceptibility to infection in diabetic patients. Therefore, inhibiting the M2 polarization of microglia could potentially reduce the susceptibility to infection in patients with diabetes.

## Figures and Tables

**Figure 1 biomedicines-13-02049-f001:**
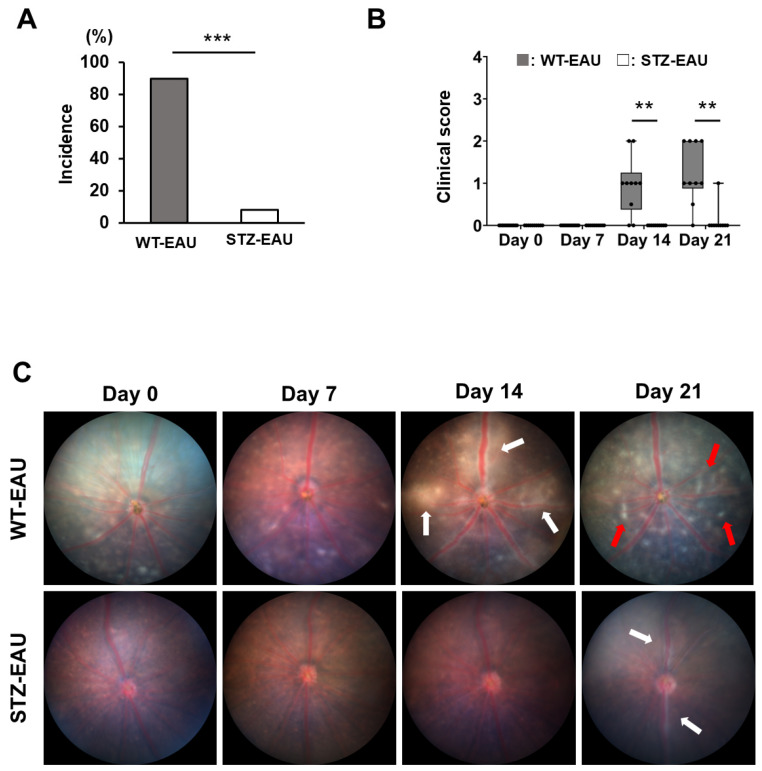
Effect of diabetes on EAU severity. Mice were subjected to funduscopy at 0, 7, 14, and 21 days post-immunization to assess EAU severity. (**A**) Incidence of EAU in WT-EAU and STZ-EAU mice, as determined by funduscopy at 21 days post-immunization. Samples with an EAU score of 0.5 or higher were considered to have developed EAU (n = 10–12 eyes per group). (**B**) EAU scores determined by funduscopy at 0, 7, 14, and 21 days post-immunization (n = 10–12 eyes per group). (**C**) Representative fundus images at each time point. Retinal vasculitis and retinal exudates are indicated by white and red arrows, respectively. Boxplots show medians with interquartile ranges (IQRs), and each dot represents an individual sample score. Statistical analysis was performed using Fisher’s exact test for incidence and the Wilcoxon rank-sum test for EAU scores. ** *p* < 0.01; *** *p* < 0.001.

**Figure 2 biomedicines-13-02049-f002:**
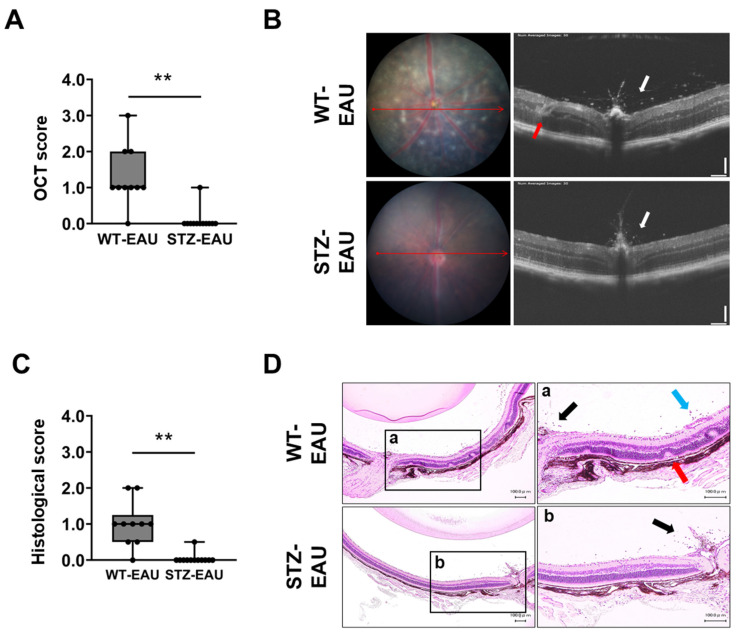
Evaluation of EAU severity by OCT and histopathology. The severity of EAU was evaluated in each mouse by OCT at 21 days post-immunization. Eyes were enucleated from each mouse under anesthesia at 21 days post-immunization, and paraffin sections were prepared and stained with H&E for histopathological evaluation. (**A**) EAU scores determined by OCT at 21 days post-immunization (n = 10–12 eyes per group). (**B**) Representative OCT images. Infiltrating cells near the optic nerve head and disrupted retinal lamination are indicated by white and red arrows, respectively. (**C**) EAU scores determined by histopathology at 21 days post-immunization (n = 10–12 eyes per group). (**D**) Representative images of H&E-stained histological sections. The left panels show low-magnification images, and the right panels show high-magnification images of the boxed regions labeled a (WT-EAU) and b (STZ-EAU) in the left panels. Black arrows indicate infiltrating cells near the optic nerve head, blue arrows indicate cells infiltrating from blood vessels, and red arrows indicate the destruction of the retinal layer structure. Scale bars: 100 µm. (**A**,**C**) Boxplots show medians with IQRs, and each dot represents an individual sample score. Statistical analysis was performed using the Wilcoxon rank-sum test. ** *p* < 0.01.

**Figure 3 biomedicines-13-02049-f003:**
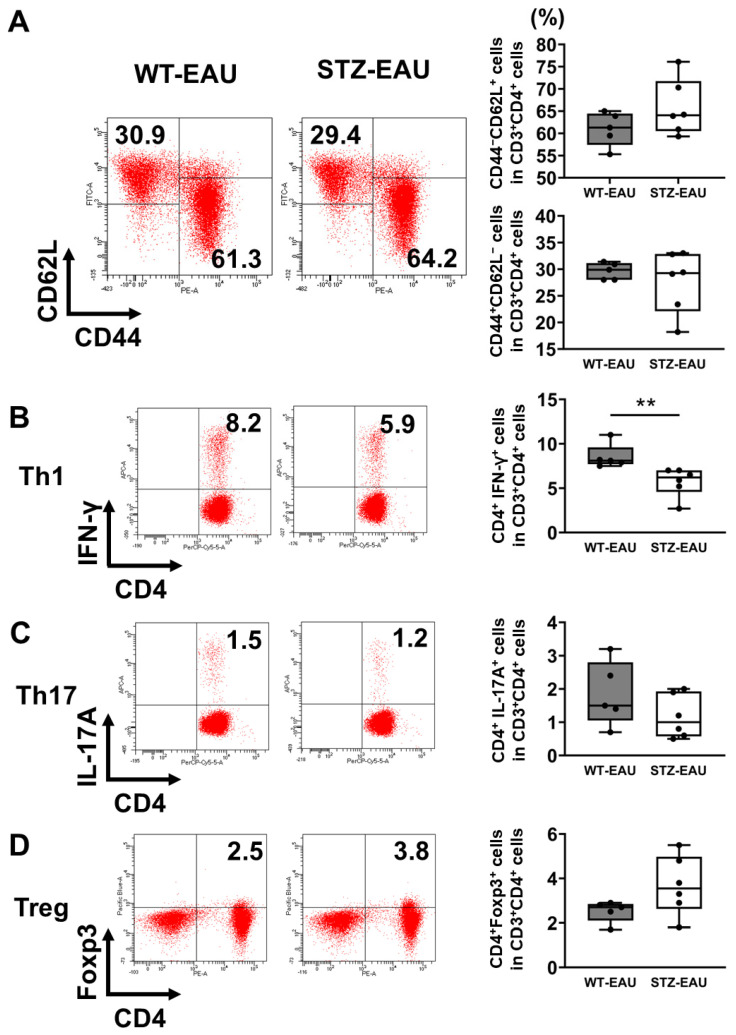
Impact of diabetes on T cell subsets involved in autoimmune uveitis. Flow cytometric analysis of T cell activation and subset distribution in CD3^+^CD4^+^ cells from mouse spleens at 21 days post-immunization. (**A**) T cell activation was assessed based on CD44 and CD62L expression. (**B**) Th1 cells were defined as IFN-γ^+^ cells. (**C**) Th17 cells were defined as IL-17A^+^ cells. (**D**) Tregs were defined as Foxp3^+^ cells. The left panels show representative data from individual mice and boxplots show medians with IQRs; each dot represents an individual positive cell percentage (n = 5–6 mice per group). Statistical analysis was performed using the Wilcoxon rank-sum test. ** *p* < 0.01.

**Figure 4 biomedicines-13-02049-f004:**
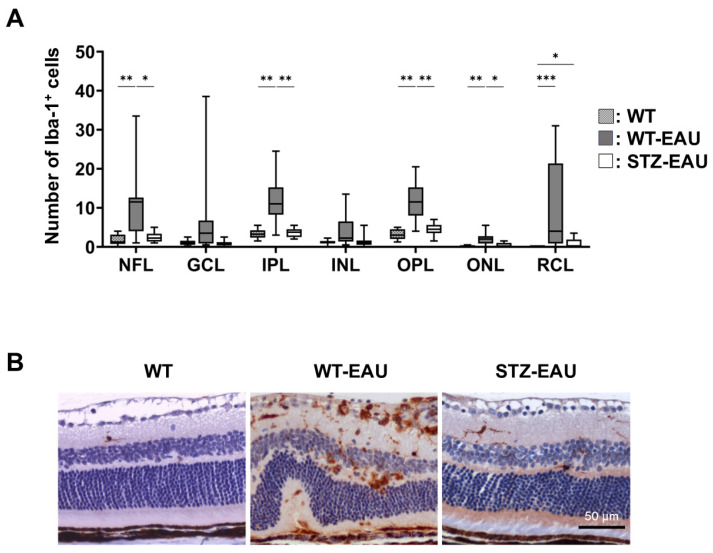
Impact of diabetes on retinal microglia involved in autoimmune uveitis. Eyes were enucleated from each mouse under anesthesia at 21 days post-immunization, and retinal sections were stained for Iba1 to quantify microglia in each retinal layer. (**A**) Box plot showing the number of retinal microglia in each retinal layer for each group. NFL, nerve fiber layer, GCL, ganglion cell layer, IPL, inner plexiform layer, INL, inner nuclear layer, OPL, outer plexiform layer, ONL, outer nuclear layer, RCL, rod and cone layer. The Steel–Dwass test was used to compare the number of cells among retinal layers. Boxplots show medians with IQRs. * *p* < 0.05; ** *p* < 0.01; *** *p* < 0.001. (**B**) Representative images of immunohistochemical staining of each group. Scale bars: 50 µm.

**Figure 5 biomedicines-13-02049-f005:**
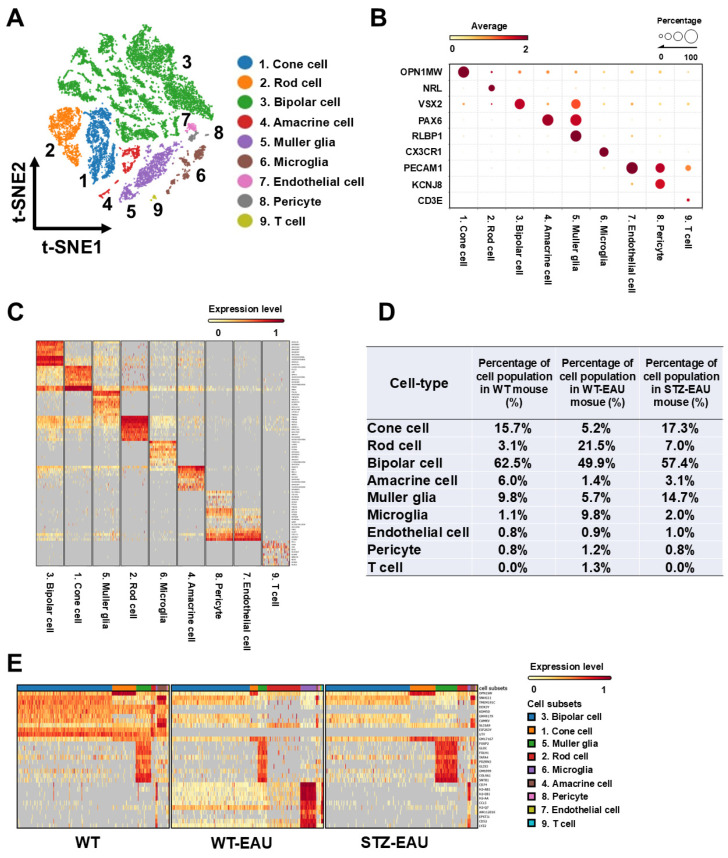
Transcriptome atlas of diabetic retinal cells in the autoimmune uveitis model. Retinas were collected from 5 mice per group (WT, WT-EAU, and STZ-EAU) at 21 days post-immunization to induce EAU. Rod cells were depleted from the pooled retinal cells of each group using magnetic bead sorting, followed by the construction of single-cell RNA sequencing libraries. The obtained data were analyzed. (**A**) Retinal cell clusters identified by cluster analysis shown in a t-SNE plot of retinal cell scRNA-seq data. (**B**) Bubble heatmap showing the proportion of retinal cell-specific gene expression in each cluster. (**C**) Heatmap showing the top genes characterizing each cluster. (**D**) Table showing the proportion of each cluster in the groups. (**E**) Heatmap showing the expression of top genes characterizing each group within each retinal cluster.

**Figure 6 biomedicines-13-02049-f006:**
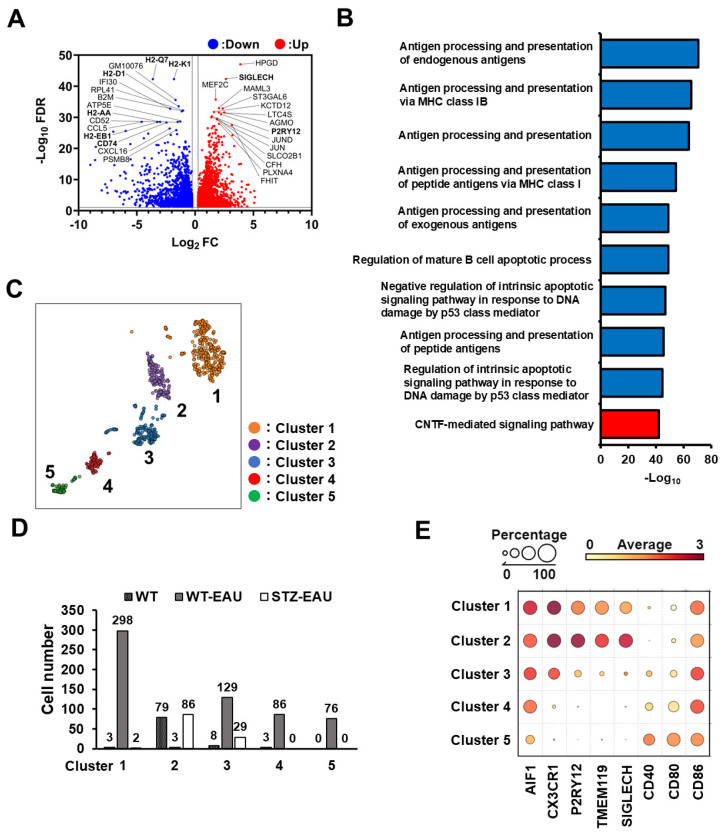
M2 polarization of retinal microglia in diabetic mice with autoimmune uveitis. (**A**) Volcano plot showing DEGs in retinal microglia between the STZ-EAU and WT-EAU groups. (**B**) Bar plot showing Gene Ontology Biological Process (GOBP) terms enriched in retinal microglia DEGs between the STZ-EAU and WT-EAU groups. Red bars indicate upregulated GOBP terms, and blue bars indicate downregulated GOBP terms. (**C**) Five sub-clusters identified by cluster analysis of the microglia cluster. (**D**) Number of cells in microglia sub-clusters 1–5 from the WT, WT-EAU, and STZ-EAU groups. (**E**) Bubble heatmap showing the expression ratio and intensity of microglia-specific genes and costimulatory molecules in sub-clusters 1–5. (**F**,**G**) Expression intensity and distribution of M1 (*CD68*, *CD74*, and *IL1B*) and M2 (*MSR1*, *CD163*, and *MRC1*) marker genes in the microglia cluster are shown in the t-SNE plot. Box plots showing the expression levels of M1 and M2 marker genes in each group within sub-cluster 3. The statistical significance of differences between groups was determined using the Steel–Dwass test. ** *p* < 0.01, *** *p* < 0.001, **** *p* < 0.0001. (**H**) GOBP terms associated with DEGs in sub-cluster 3 in the comparison of the STZ-EAU and WT-EAU groups. Red bars indicate upregulated GOBP terms, while blue bars represent downregulated GOBP terms.

## Data Availability

The scRNA-seq data sets presented in the study are openly available in the DNA Data Bank of Japan (DDBJ) under the bioproject accession number PRJDB19651 (https://ddbj.nig.ac.jp/search/entry/bioproject/PRJDB19651). All other data supporting the reported results are contained within the article and its Appendix A, and are available from the corresponding author upon request.

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
