# Peer review of "Enhanced M2 Polarization of Retinal Microglia in Streptozotocin-Induced Diabetic Mice upon Autoimmune Stimulationâ€"

_biomedicines, 2025, doi:10.3390/biomedicines13092049_

Round 1
Reviewer 1 Report (Previous Reviewer 4)
Comments and Suggestions for Authors
Manuscript can be accepted for publication
Author Response
We appreciate your time and thoughtful review.
Reviewer 2 Report (New Reviewer)
Comments and Suggestions for Authors
A major revision of the manuscript is necessary before the manuscript is recommended for publication in Biomedicines.

The English could be improved to more clearly express the research.
Author Response
Comments and Suggestions for Authors
A major revision of the manuscript is necessary before the manuscript is recommended for publication in Biomedicines.
Responses:
We appreciate your suggestion regarding the need for English editing of our manuscript. We have carefully reviewed the entire text of the submitted manuscript and revised it to ensure greater clarity of expression. The revised portions are indicated in red text in the resubmitted version.
In considering this resubmission, we explored the possibility of using the English and figure editing services provided by MDPI. However, due to regulations at our institution, payments to designated service providers require the completion of internal administrative procedures, which typically take approximately three weeks. Given the revision period of 10 days, we were unfortunately unable to make use of these services.
We are confident that the present revisions have improved our manuscript and made it more comprehensible for readers.
Reviewer 3 Report (New Reviewer)
Comments and Suggestions for Authors
This is an interesting manuscript. Well-written. One main thing that is missing is the group of only STZ. Furthermore, as a minor point, timeline rationale e.g. post-STZ or post-EAU should be presented clearly. It is related with disease progression. Several data lost SD values in images for example, Figure 6. It might need adjustment of the figure for technical issues.
Author Response
Comments and Suggestions for Authors
This is an interesting manuscript. Well-written. One main thing that is missing is the group of only STZ. Furthermore, as a minor point, timeline rationale e.g. post-STZ or post-EAU should be presented clearly. It is related with disease progression. Several data lost SD values in images for example, Figure 6. It might need adjustment of the figure for technical issues.
Response:
We sincerely appreciate your positive evaluation of our manuscript. As you correctly noted, our study design did not include an STZ-only mouse group. A comparative analysis between STZ mice and WT mice using scRNA-seq has already been performed by Licheng Sun et al. (Reference 26). For this reason, we did not conduct an analysis of STZ-only mice in this study.
In addition, we have revised the RESULTS section to make the timing of sample evaluation for each experimental group more explicit. In many studies, the experimental design is divided into disease and treatment groups. However, in our study, we combined the EAU group, which represents the disease condition, with the diabetic (STZ) group, which is typically treated as a disease group in conventional research, to create the STZ-EAU group and thereby investigate the impact of diabetes on the development of autoimmune disease. We recognize that this approach may potentially cause confusion for readers in understanding the design of our study. Therefore, we have also refined the English expression throughout the manuscript to further improve clarity and facilitate the reader’s understanding. These revisions are highlighted in red text in the revised manuscript for your reference.
Regarding Figure 6D, standard deviations are not shown in the bar graph because scRNA-seq data were obtained only once. However, to enhance data robustness, we pooled retinal samples from five mice per group for scRNA-seq. This methodology is described in the legend of Figure 5, but we have also added it to the Materials and Methods section under Single-Cell RNA Sequencing (scRNA-seq), Retinal Single-Cell Preparation.
We are confident that these revisions have improved the quality and clarity of our manuscript.
Round 2
Reviewer 2 Report (New Reviewer)
Comments and Suggestions for Authors
This manuscript is recommended for acceptance.
This manuscript is a resubmission of an earlier submission. The following is a list of the peer review reports and author responses from that submission.
Round 1
Reviewer 1 Report
Comments and Suggestions for Authors
Most of all, the manuscript did not substantially change despite the argumentation in the cover. In addition, the new experiments requested by the revision have not been performed.
Previous comments:
Almost impossible to read and mostly to understand this paper.
First, no chance to understand its aim .
The paper starts with the fact that EAU was suppressed in Akita mice with a spontaneous mutation in the insulin 2 gene resulting in severe insulin-dependent diabetes. In this line, the authors demonstrated that EAU is suppressed in mice with streptozotocin-induced diabetes (STZ-mice) which do not have genetic mutations.
- Which is the added value of this finding in respect to Akita?
- How diabetes occurrence after STZ has been assessed. Methods and results are lacking
- Is diabetes occurrence complicated by DR?
- Why environmental diabetes suppresses EAU occurrence? The likely explanation is that STZ-induced diabetes suppresses the development of EAU, “possibly by modulating microglial activation and polarization towards an M2 phenotype, leading to decreased antigen presentation and potentially reduced recruitment of inflammatory cells to the retina”. Please, explain it comprehensibly mostly in respect to the importance of this finding .
- STZ-mice without EAU occurrence irrespectively of EAU induction in comparison with EAU induction alone? Rather difficult to understand.
- To which aim? To understand the mechanisms underlying EAU? Indeed, difficult to decipher even because of the difficulty to understand the English language.
- In Akita, tha lack of EAU was found to depend on the fact that Th1 differentiation was significantly lower than in WT-EAU mice, while differentiation into Th17 and Treg cells was preserved. And………..?
- What does it mean?
- How the present findings in STZ mice can be used at the translational level?
- Microglial impairment in diabetes has been well established but I cannot see the link with EAU
- How this knowledge can help at the translational level?
Reviewer 2 Report
Comments and Suggestions for Authors
The manuscript entitled “Enhanced M2 polarization of retinal microglia in streptozotocin (STZ)-induced diabetic mice upon autoimmune stimulation” shows interesting information for the understanding of the effect of diabetic conditions on the activities of immune cells, especially, in the retinal tissue. The authors found that experimental autoimmune uveoretinitis (EAU) was suppressed in STZ-induced diabetic mice. It thus caused the decrease in T cell response to pathogens in diabetic patients.
The authors prepared the manuscript well and also showed good results for supporting their hypotheses.
In this current manuscript, the authors pointed out the benefits from the study of M2 polarization of retinal microglia in STZ-induced diabetic mice upon autoimmune stimulation. Furthermore, it has been improved following the suggestions from the previous review. Therefore, this manuscript could be accepted for publishing in Biomedicines as the present form.
Author Response
Comments
The manuscript entitled “Enhanced M2 polarization of retinal microglia in streptozotocin (STZ)-induced diabetic mice upon autoimmune stimulation” shows interesting information for the understanding of the effect of diabetic conditions on the activities of immune cells, especially, in the retinal tissue. The authors found that experimental autoimmune uveoretinitis (EAU) was suppressed in STZ-induced diabetic mice. It thus caused the decrease in T cell response to pathogens in diabetic patients.
The authors prepared the manuscript well and also showed good results for supporting their hypotheses.
In this current manuscript, the authors pointed out the benefits from the study of M2 polarization of retinal microglia in STZ-induced diabetic mice upon autoimmune stimulation. Furthermore, it has been improved following the suggestions from the previous review. Therefore, this manuscript could be accepted for publishing in Biomedicines as the present form.
Response
We appreciate you taking the time to review our resubmitted manuscript and for your valuable comments. We hope that our findings will contribute to a better understanding of diabetes in relation to intraocular inflammatory diseases.
Reviewer 3 Report
Comments and Suggestions for Authors
The work presented for review by Yoshiaki Nishio and co-authors demonstrates M2 polarization of retinal microglia in mice with modeled diabetes and autoimmune stimulation. The study of microglial polarization is currently a trending research area, as confirmed by the PubMed database. Furthermore, according to WHO data, the number of patients with type 1 and type 2 diabetes worldwide increases annually. Both of these factors indicate the undeniable relevance of the obtained data. Upon reading, only minor comments were formulated, the list of which is provided below:
- It appears that there is an error in the glucose concentration units, as volume measurements are presented in both the numerator and denominator. Additionally, it is necessary to specify how and with what frequency blood was collected for analysis, as well as what equipment was used for the measurements.
- Why did the authors use the Wilcoxon test instead of the Mann-Whitney criterion? Which test was used to determine sample normality? If the authors employ non-parametric tests (e.g., Wilcoxon), how can they use mean values and SD instead of median and interquartile range when indicating values in the text and constructing diagrams (e.g., Figure 1B, C and other diagrams)? If the same mice were used for the data in Figure 1, then the data should be presented for each individual mouse over time. If these were different animals, then what is the actual number of animals selected for the experiments?
Author Response
Comments and Suggestions
- It appears that there is an error in the glucose concentration units, as volume measurements are presented in both the numerator and denominator. Additionally, it is necessary to specify how and with what frequency blood was collected for analysis, as well as what equipment was used for the measurements.
Response
We are grateful for your numerous insightful comments on our manuscript. We have summarized our responses to your points below. All revisions in the manuscript are highlighted in red.
We appreciate you pointing out the error in the blood glucose unit. We also apologize for the insufficient description of the blood glucose measurement method. We have corrected the blood glucose unit to mg/dL and added details regarding the blood collection method and measurement frequency to the "Creation of STZ-induced diabetic mice" section within MATERIALS AND METHODS.
- Why did the authors use the Wilcoxon test instead of the Mann-Whitney criterion? Which test was used to determine sample normality? If the authors employ non-parametric tests (e.g., Wilcoxon), how can they use mean values and SD instead of median and interquartile range when indicating values in the text and constructing diagrams (e.g., Figure 1B, C and other diagrams)? If the same mice were used for the data in Figure 1, then the data should be presented for each individual mouse over time. If these were different animals, then what is the actual number of animals selected for the experiments?
Responses
We understand that the Wilcoxon rank-sum test and the Mann-Whitney U test are equivalent non-parametric statistical methods. The JMP software, which we used for statistical analysis, supports only the Wilcoxon rank-sum test for two-group non-parametric comparisons. Therefore, we consistently employed the Wilcoxon rank-sum test for all two-group comparisons in this manuscript.
For this study, the sample size for each experimental group was small (N=5 or 6 animals). Given this small sample size, we determined that performing normality testing using the Shapiro-Wilk test would not be appropriate due to insufficient statistical power. Consequently, we consistently used non-parametric statistical analysis throughout this study.
We appreciate you pointing out the issue with our data presentation. We have revised the data in Figure 1B, Figure 2A and C, and Figure 3A-D to box plots, and accordingly, the numerical values in the text have been updated to represent the median and interquartile range.
Figure 1B illustrates the time-course EAU scores based on funduscopic examination, derived from 10 eyes of 5 WT-EAU mice and 12 eyes of 6 STZ-EAU mice. Although these data were obtained from the same animals over time, we opted for box plots as we determined that a scatter plot with connecting lines for each sample would compromise data clarity. The exact number of animals (n=5-6 mice per group) used for each experiment is explicitly stated in the respective Figure legends.
Reviewer 4 Report
Comments and Suggestions for Authors
In the article “Enhanced M2 polarization of retinal microglia in streptozotocin-induced diabetic mice upon autoimmune stimulation”, Nishio et al. present the impact of diabetes on the development of experimental autoimmune uveoretinitis (EAU) and the activation of microglia in the retina. The authors demonstrate that mice with streptozotocin-induced diabetes exhibit reduced severity of EAU, which is associated with a decreased number of Th1 cells and a shift in microglial polarization toward an anti-inflammatory M2 phenotype. The findings suggest a potential link between the diabetic environment and increased susceptibility to infections due to alterations in the immune response. This work is of significant interest for understanding the mechanisms of immune dysregulation in diabetes and may have therapeutic implications.
- The “Materials and Methods” section lacks specific information on the anesthetic used for the animals. Please include details on the drug and dosage.
- Please provide a detailed protocol for deparaffinization and hematoxylin and eosin staining.
- Kindly include the number of animals in each experimental group.
- Was verification of diabetes level performed using other metabolic markers besides blood glucose analysis?
- Was a quantitative analysis of the M1/M2 microglia ratio performed at different stages of the disease?
- In your study, you present CD68, CD74, and IL1B as M1 markers and MSR1, CD163, and MRC1 as M2 markers. However, CD68, CD74, MSR1, and MRC1 are markers not specific to microglia alone but at least also to macrophages and some other cell types. Perhaps you should reconsider the terminology and use “microglia/macrophages” instead of just “microglia,” or revise the interpretation of your study results.
The results obtained are of significant importance for understanding the underlying mechanisms of the immune response in diabetes and hold promise for the development of therapies for this condition. In its current form, however, the manuscript is not suitable for publication due to major concerns. After substantial revisions, the article may be considered for recommendation for publication.
Author Response
Comments and Suggestions for Authors
1. The “Materials and Methods” section lacks specific information on the anesthetic used for the animals. Please include details on the drug and dosage.
Response
We are grateful for your insightful comments, which have greatly improved our manuscript. We have summarized the corrections and our responses to your comments below. All revisions in the manuscript are highlighted in red.
We appreciate you pointing out the missing details about the anesthesia conditions in our animal experiments.
We've added the anesthesia specifics under "Animals" in the Materials and Methods section. Additionally, we've included the euthanasia conditions in the "Flow cytometry" and "Retinal Single-Cell Preparation" subsections within "Single-Cell RNA sequencing (scRNA-seq)."
2. Please provide a detailed protocol for deparaffinization and hematoxylin and eosin staining.
Response
We apologize for the insufficient description of the deparaffinization and H&E staining methods. We have added these details to the "Histological and Immunohistochemical Analysis" section within the Materials and Methods.
3. Kindly include the number of animals in each experimental group.
Response
We appreciate your comment regarding the number of experimental animals used in this study. The precise number of animals utilized in each experiment is detailed in the respective Figure legends.
4. Was verification of diabetes level performed using other metabolic markers besides blood glucose analysis?
Response
We apologize, the evaluation of diabetic mice in this study was limited to the time-course observation of blood glucose levels and body weight, the results of which are presented in Supplemental Figure 1. Consequently, other metabolic markers were not assessed.
5. Was a quantitative analysis of the M1/M2 microglia ratio performed at different stages of the disease?
Response
In this study, M1/M2 microglial analysis was performed exclusively using scRNA-seq data, samples collected 21 days post-EAU immunization. Therefore, we did not investigate the ratios of M1/M2 microglia at different stages of the disease.
6. In your study, you present CD68, CD74, and IL1B as M1 markers and MSR1, CD163, and MRC1 as M2 markers. However, CD68, CD74, MSR1, and MRC1 are markers not specific to microglia alone but at least also to macrophages and some other cell types. Perhaps you should reconsider the terminology and use “microglia/macrophages” instead of just “microglia,” or revise the interpretation of your study results.
Response
We apologize for the insufficient explanation regarding the microglial analysis data.
In Figure 6, we performed sub-cluster analysis on the microglial cluster identified in Figure 5. Within the resulting sub-clusters 1-5, sub-clusters 1, 2, and 3 all expressed microglia-specific markers CX3CR1, P2RY12, TMEM119, and SIGLECH, leading us to identify them as microglial populations. Conversely, sub-clusters 4 and 5, which did not express these microglia-specific markers, are considered to be populations of infiltrating macrophages.
In the box plots of Figure 6F and 6G, we compared the expression of M1 and M2 markers within sub-cluster 3, which represents a population of activated microglia, across each sample group. Therefore, the M1/M2 marker data presented in this manuscript exclusively refer to sub-cluster 3, the activated microglial population. To enhance the clarity of the figures, we have revised the box plots in Figure 6F and 6G, as well as Supplementary Figure 5.